# Glycoproteomic Analysis Reveals Aberrant Expression of Complement C9 and Fibronectin in the Plasma of Patients with Colorectal Cancer

**DOI:** 10.3390/proteomes8030026

**Published:** 2020-09-22

**Authors:** Juthamard Chantaraamporn, Voraratt Champattanachai, Amnart Khongmanee, Chris Verathamjamras, Naiyarat Prasongsook, Kanokwan Mingkwan, Virat Luevisadpibul, Somchai Chutipongtanate, Jisnuson Svasti

**Affiliations:** 1Laboratory of Biochemistry, Chulabhorn Research Institute, Bangkok 10210, Thailand; juthamard@cri.or.th (J.C.); chris@cri.or.th (C.V.); jisnuson@cri.or.th (J.S.); 2Applied Biological Science Program, Chulabhorn Graduate Institute, Chulabhorn Royal Academy, Bangkok 10210, Thailand; 3Translational Research Unit, Chulabhorn Research Institute, Bangkok 10210, Thailand; amnart@cri.or.th; 4Division of Medical Oncology, Department of Medicine, Faculty of Medicine, Phramongkutklao Hospital, Ratchathewi, Bangkok 10400, Thailand; naiyarat_p@yahoo.co.th; 5Division of Surgery, Sapphasitthiprasong Hospital, Ubon Ratchathani 34000, Thailand; cancer_sappasit@hotmail.com; 6Division of Information and Technology, Ubonrak Thonburi Hospital, Ubon Ratchathani 34000, Thailand; viratlue@gmail.com; 7Pediatric Translational Research Unit, Department of Pediatrics, Faculty of Medicine Ramathibodi Hospital, Mahidol University, Bangkok 10400, Thailand; schuti.rama@gmail.com; 8Department of Clinical Epidemiology and Biostatistics, Faculty of Medicine Ramathibodi Hospital, Mahidol University, Bangkok 10400, Thailand

**Keywords:** biomarker, colorectal cancer, complement C9, fibronectin, label-free quantitative proteomics, wheat germ agglutinin

## Abstract

Colorectal cancer (CRC) is a major cause of cancer mortality. Currently used CRC biomarkers provide insufficient sensitivity and specificity; therefore, novel biomarkers are needed to improve the CRC detection. Label-free quantitative proteomics were used to identify and compare glycoproteins, enriched by wheat germ agglutinin, from plasma of CRC patients and age-matched healthy controls. Among 189 identified glycoproteins, the levels of 7 and 15 glycoproteins were significantly altered in the non-metastatic and metastatic CRC groups, respectively. Protein-protein interaction analysis revealed that they were predominantly involved in immune responses, complement pathways, wound healing and coagulation. Of these, the levels of complement C9 (C9) was increased and fibronectin (FN1) was decreased in both CRC states in comparison to those of the healthy controls. Moreover, their levels detected by immunoblotting were validated in another independent cohort and the results were consistent with in the study cohort. Combination of CEA, a commercial CRC biomarker, with C9 and FN1 showed better diagnostic performance. Interestingly, predominant glycoforms associated with acetylneuraminic acid were obviously detected in alpha-2 macroglobulin, haptoglobin, alpha-1-acid glycoprotein 1, and complement C4-A of CRC patient groups. This glycoproteomic approach provides invaluable information of plasma proteome profiles of CRC patients and identification of CRC biomarker candidates.

## 1. Introduction

Colorectal cancer (CRC) ranks as the third most common cancer accounting for over 1.8 million cases per year based on the global cancer statistics of 2018 [1]. CRC is curable if detected at early stages, reducing CRC mortality. Colonoscopy is a gold standard method for detecting CRC; however, people tend to avoid undergoing this technique because of its invasiveness [2]. Currently, carcinoembryogenic antigen (CEA) is the only blood-based biomarker approved for the clinical use to monitor CRC recurrence [3]. However, CEA is not entirely satisfactory for routine clinical analysis in terms of sensitivity and specificity, so that its level may be found to be altered in benign and inflammatory conditions [3,4]. Finding novel specific and sensitive biomarkers from liquid biopsy specimens such as plasma and serum would be an ideal and relatively non-invasive way for CRC screening and detection.

Circulating blood proteins are invaluable resources which may reflect pathological conditions, including cancers. However, seeking potential protein biomarkers in serum/plasma is difficult because of the several proteins present and in addition, potential biomarkers may be overshadowed by high abundant proteins, such as albumin. To enhance the detection of low abundant tumor-specific biomarkers, sub-proteomes that target a specific subset of serum/plasma glycoproteins was investigated [5]. Focusing on the glycoproteome has received a wide attention in cancer research since altered protein glycosylation has been shown a correlation with the development of cancers [6]. Many serum/plasma glycoproteins have been commercially applied for screening cancer, including prostate-specific antigen (PSA) in prostate cancer, cancer antigen 15-3 (CA15-3) in breast cancer, and cancer antigen 125 (CA-125) in ovarian cancer [4,7,8].

Lectins, carbohydrate-binding proteins, have become one of the most exploited tools for studying glycoproteins in biological samples, since they can bind reversibly to specific glycan structures on glycoproteins. They have been used for purification and detection of glycoproteins that show aberrant expression in many cancers such as breast cancer, prostate cancer, lung cancer, hepatocellular carcinoma, pancreatic cancer, and CRC [9,10,11]. Lectin histochemistry demonstrated that the binding of wheat germ agglutinin (WGA), a lectin binding protein that specifically binds to GlcNAc/sialic acid residues, was increased in CRC tissues, suggesting that glycoproteins bearing specific sugars may be associated with malignant progression [12].

In the present study, glycoproteins in plasma samples from two states of CRC patients (non-metastatic and metastatic CRC), as well as healthy controls, were enriched using WGA affinity chromatography and the enriched glycoproteins were analyzed by liquid chromatography coupled with tandem mass spectrometer (LC-MS/MS). The differential expression levels of plasma glycoproteins were compared and protein-protein interactions of those proteins were analyzed. The candidate protein was confirmed by immunoblotting in the plasma samples of the study cohort and in the serum samples of an independent (validation) cohort. The results are statistically analyzed for their potential in detection of CRC in terms of sensitivity and specificity.

## 2. Materials and Methods

### 2.1. Patients and Specimens

In the study cohort, samples (*n* = 30) were EDTA-plasma collected from 20 patients diagnosed with CRC (10 patients with non-metastatic states and 10 patients with metastatic states) and 10 aged-match healthy controls. All plasma samples were obtained from the Sapphasitthiprasong Hospital (Ubon Ratchathani, Thailand) and approved by the Institute Review Board (IRB) of Ramathibodi Hospital (protocol ID #03-58-68). In the validation cohort, samples (*n* = 26) were serum collected from 14 patients with CRC (6 patients with non-metastatic states and seven patients with metastatic states) and 13 aged = match healthy controls. All serum samples were obtained from Phramongkutklao Hospital and approved by the Institutional Review Board of the Royal Thai Army Medical Department, Thailand (protocol ID #S012h/56). The CRC patient samples were collected before surgery or chemotherapy while the healthy control samples were obtained from yearly check-up participants who gave informed consent at the hospitals. Aliquots of samples were stored at −80 °C until use for further analysis. The characteristics of all samples are summarized in Table 1.

### 2.2. WGA-Bound Glycoproteins Enrichment

Crude plasma samples in the study cohort were divided in three groups (10 samples/group; healthy controls, non-metastatic and metastatic CRC patients). Protein concentration of each sample was measured by the Bradford assay (Bio-Rad Laboratories, Hercules, CA, USA). Equal amounts of protein from individual samples of each group were pooled and served as a representative of each group. The glycoprotein enrichment was performed from the pooled sample of each group using wheat germ agglutinin (WGA) kit (Pierce, Thermo Scientific, Rockford, IL, USA). Prior to starting, the pooled plasma samples were centrifuged through Costar Spin-X 0.22 µm centrifuge filters (Corning Incorporated Life Sciences, Corning, NY, USA) for 1 min at 16,000× *g*. EDTA in the clear supernatant was removed using Bio-Spin centrifugal devices (Bio-Rad Laboratories, Hercules, CA, USA). EDTA-depleted plasma samples (1 mg) were applied in the WGA kit, following the manufacturer’s recommendations. After enrichment, the WGA-bound and unbound fractions were supplemented with 1% protease inhibitor cocktail (Sigma-Aldrich, Saint Louis, MO, USA). Protein concentration of each fraction was measured by the Bradford assay. All samples were stored at −80 °C until processing.

### 2.3. Assessment of Total Plasma Protein Levels

The pooled samples of three groups received from crude plasma, WGA-bound and WGA-unbound plasma fractions were separated by 10% SDS-PAGE (TGX Stain-Free FastCast, Bio-Rad). The images of total proteins were captured and digitalized using a Stain-free tray with Gel Doc™ EZ Imager (Bio-Rad Laboratories, Hercules, CA, USA).

### 2.4. Sample Preparation for LC-MS/MS Analysis

The pooled samples of WGA-bound fractions were buffer-exchanged in 50 mM ammonium bicarbonate (NH_4_HCO_3_) using a Bio-spin centrifugal device. Ten micrograms of WGA-bound glycoproteins in 50 mM NH_4_HCO_3_ were reduced with 10 mM dithiothreitol (DTT) at 95 °C for 5 min and alkylated with 20 mM iodoacetamide (IAA) in the dark at RT for 30 min. The proteins were digested with trypsin (Promega, Madison, WI, USA) at ratio of trypsin: Protein = 1:50 *w*/*w* at 37 °C overnight. The digest was stopped by adding 100% formic acid (FA) to a final concentration of 1% *v*/*v*. The peptides were dried by SpeedVac (Labconco, Kansas City, MO, USA) and cleaned up using TopTip C-18 (Glygen Corporation, Columbia, MD, USA). The cleaned-up peptide samples were immediately dried using SpeedVac and stored at −20 °C. Prior to LC-MS/MS analysis, the dried-peptides were resuspended in 0.1% FA.

### 2.5. LC-MS/MS Analysis

Mass spectral data were acquired using LC-MS/MS, an Orbitrap Elite™ Hybrid Ion Trap-Orbitrap Mass Spectrometer coupled to an UltiMate™ 3000 RSLCnano System (Thermo Fisher, San Jose, CA, USA). The mass spectrometer was operated using XCalibur software (Thermo Fisher, San Jose, CA, USA). The mass range for MS scans was set to m/z 250–2000 at resolution 120,000 using FTMS analyzer. The MS/MS spectrum were selected from top seventh most abundant parent ions for each MS scan. The normalized collision energy for each MS/MS (CID) event using ITMS analyzer was set to 35%. Each peptide sample equivalent to 200 ng of undigested protein was subjected to EASY- Spray™ HPLC column (P/N ES802A, Thermo Fisher, San Jose, CA, USA) using an UltiMate™ 3000 RSLCnano System. The samples were loaded to EASY-SPRAY™ column using 2% acetonitrile (ACN) aqueous solution containing 0.1% formic acid. The separation was attained using a solvent gradient ramping from 2 to 40% aqueous ACN (0.1% formic acid) over 60 min at a flow rate of 300 nL/min. Each sample was analyzed in triplicate, providing three technical replicates per sample.

### 2.6. Label-Free Relative Quantification and Identification of Proteins

Label-free relative quantification was performed using Progenesis QI v3.1 (Nonlinear Dynamics, Milford, MA, USA). The comparison of spectra among multiple LC-MS runs of three sample groups provided quantitative measurement of peptides. Thus, in order to match peptide ions across different LC-MS runs, all nine LC-MS runs in the experiment (triplicate runs of non-metastatic CRC, metastatic CRC, and healthy groups) were aligned together based on the LC retention time (RT) and *m*/*z* ratio. According to Progenesis QI software, the run that showed ‘least difference’ in peptide ions from all nine LC-MS runs in the data was automatically selected to be the reference run. Retention time (RT) and *m*/*z* values of the other runs were aligned by the reference run. Then the raw LC-MS peak intensity for each individual spectrum was transformed to a normalized LC-MS peak intensity, based on LC-MS peak intensity of the reference run. Parameters were used to filter the data before exporting the MS/MS spectra files to Byonic software v3.3 (Protein Metrics, Cupertino, CA, USA) for peptide/protein identification including mass peaks acquired in the 250–2000 *m*/*z* range and the retention time limited to 0 to 70 min with charge states from +1 to +6.

MS/MS spectra obtained were searched against the Uniprot Human Proteome Database (canonical sequences of 20,416 entries, 16 March 2019). The search was performed by setting parameters as follows: enzyme was set as trypsin with a maximum of two missed cleavages allowed; precursor mass tolerance = 8 ppm; fragment mass tolerance = 0.3 Da; fragmentation type was set as CID low energy; carbamidomethyl of cysteine was set as fixed modification; dethiomethyl of cysteine, deamidation of asparagine, conversion of glutamine to pyroglutamic acid at N-terminal, conversion of glutamate to pyroglutamine at N-terminal, amino-loss at N-terminal of cysteine, and acetylation at N-terminal of proteins were set for variable modifications. A 1% false discovery rate (FDR) was applied to exclude false positive identifications.

The peptide search data were imported back into Progenesis QI. Then, the identified peptides were further filtered and “confident peptides” considered using the following criteria: a confidence threshold of |Log Prob| greater than 1.3; ANOVA < 0.05; selection of only unique sequence peptides with non-conflicting (non-conflicting defined as a MS/MS compound used only once to assign as one peptide); removing incorrect proteins that included reverse sequences and keratin contaminants. Lastly, only proteins containing at least two peptides were considered for relative quantification of protein. The fold changes of each protein in non-metastatic and metastatic CRC patients and healthy controls were calculated from the MS peak intensities of accepted peptides. Then, the intensities of all accepted peptides were combined and normalized to those of the healthy control group.

In addition, the MS/MS spectra of the pooled one sample of each group were search against the top 20 WGA-enriched plasma proteins with a 1.5-fold change cutoff in differential expression in non-metastatic or metastatic CRC patients compared to those of the healthy controls using Byonic software v3.3 for N-linked glycosylation and glycation. The search was performed by above setting parameters with additional modifications as follows: 148 forms of N-linked glycosylation (using combined human N-linked database from Byonic) and 15 forms of glycation modifications including (fructosyl–lysine-2H_2_O, FL-2H_2_O (+126.0317 Da), carboxy-methyllysine (CML) (+58.0055 Da), carboxyethyllysine (CEL) (+72.0211 Da), Pyrraline (+108.0211 Da)); those involving arginine-specific residues (imidazolone-B (+142.03 Da), argpyrimi-dine (+80.0262 Da), Ne-(5-hydro-5-methyl-4-imidazolon-2-yl) ornithine, MG–H1 (+54.0106 Da), Ne-(5-hydro-4-imidazolon-2-yl) ornithine G–H1(+39.9949 Da)) and those involving both lysine and arginine residues ((+39.9949 Da), 1-alkyl-2-formyl-3,4-glycosyl-pyrrole (AFGP) (+270.074 Da), Amadori product (+162.02 Da), imidazolone-A (+144.03 Da), methylglyoxal lysine dimer (MOLD) (+49.0078 Da), methylglyoxal-derived imidazolium cross-link (MODIC) (+36.0179 Da) and Crossline (+252.11 Da)).

### 2.7. Protein-Protein Interaction (PPI) Network Analysis

The Search Tool for the Retrieval of Interacting Genes/Proteins (STRING) database v11.0 (http://string-db.org/) was used to analyze the PPI of significantly expressed glycoproteins in plasma samples of CRC patients with non-metastasis and metastasis compared to those of the healthy controls (*p*-value < 0.05), and Cytoscape software v3.2 (1999 Free Software Foundation, Inc., Boston, MA, USA) was used to construct the PPI network. A network was constructed consisting of nodes and lines in which each node represents a protein and the lines represent direct interactions between proteins.

### 2.8. Immunoblot Analysis

To verify the glycoprotein observed in label-free relative quantification, immunoblot analyses of complement C9 (C9) and fibronectin (FN1) were performed. Equal amounts of protein from non-metastatic CRC, metastatic CRC, and healthy control groups of crude plasma, WGA-bound and WGA-unbound fractions were separated by 10% TGX Stain-Free FastCast. The total protein levels of each sample were visualized using Gel Doc™ EZ Imager and then transferred onto PVDF membranes. The membranes were blocked in 3% bovine serum albumin (BSA) in TBS/T for 1 h and probed with primary antibodies; anti-complement C9 (ab173302, Abcam, Cambridge, UK) and anti-fibronectin (ab32419, Abcam, Cambridge, UK) at 4 °C overnight. After washing, the membranes were incubated with an anti-rabbit secondary antibody conjugated with horseradish peroxidase (HRP) (P0217, Dako, Glostrup, Denmark) in 5% skim milk in TBS/T at RT for one hour. The signal on the membrane was visualized using WesternBright ECL detection kit (Advansta, San Jose, CA, USA) and captured by ImageQuant LAS 4000 digital imaging system (GE Healthcare, Piscataway, NJ, USA). The band intensities of C9 and FN1 were normalized to its stained total protein intensity on the gels for each sample.

### 2.9. CEA ELISA Assay

The concentrations of CEA in plasma samples were measured using commercial ELISA kits (CEA Human SimpleStep ELISA kit; ab183365, Abcam, Cambridge, UK). The assay procedure was performed according to the instructions of manufacturer.

### 2.10. Statistical Analysis

Statistical analysis of biomarkers among non-metastatic CRC, metastatic CRC and healthy control groups were performed using the non-parametric Mann-Whitney test, GraphPad Prism v6 (GraphPad Software, San Diego, CA, USA). Statistical significance was considered when *p*-value < 0.05. Receiver operator characteristic curve (ROC) and area under the ROC curves (AUC) were generated using GraphPad Prism to assess the diagnostic performance of each biomarker and determine the specificity and sensitivity of the biomarker. Binary logistic regression model was conducted with the Statistical Package for the Social Sciences, v23.0 (SPSS, IBM Corp, Armonk, NY, USA) to combine the diagnostic performance of the biomarkers and enabled calculation of the ROC curve, sensitivity, and specificity.

## 3. Results

### 3.1. Experimental Overview to Study Plasma Glycoprotein Biomarkers for CRC

The workflow for searching plasma glycoprotein biomarkers of CRC is shown in Figure 1. To reduce variation which may occur among individual samples, three groups of pooled plasma samples obtained from healthy controls, patients with non-metastatic and metastatic CRC were studied in a discovery phase (Figure 1A). Glycoproteins from pooled plasma samples of each group were enriched using WGA affinity chromatography. WGA-bound plasma proteins were in-solution digested by trypsin and injected to a nano-LC coupled with Orbitrap mass spectrometer (LC-MS/MS). Samples were analyzed from the three technical replicates. Then relative quantification of WGA-bound plasma proteins among three groups (three technical replicates/group) was performed by label free quantification using Progenesis QI and the proteins were identified by Byonic search engine. After statistical analysis, potential glycoprotein candidates were selected to confirm their expression levels in the same plasma samples using immunoblot assay. The selected biomarker candidates were also conformed in individual plasma samples in the study cohort (Figure 1B) as well as validated in another (validation) cohort (Figure 1C) using immunoblot assay. The level of CEA, a routinely used clinical biomarker for CRC, was also determined in individual samples of both cohorts by ELISA. Then diagnostic accuracy parameters, including ROC, AUC, sensitivity and specificity, were determined for the candidate glycoproteins and CEA.

### 3.2. Enrichment of Plasma Glycoproteins Using WGA

Protein samples from crude plasma, WGA-bound and WGA-unbound plasma fractions of three pooled plasma groups: non-metastatic CRC, metastatic CRC, and healthy controls, were resolved by 10% stain-free SDS-PAGE. The results revealed a distinct pattern of proteins among three types of sample (Figure 2). High abundant proteins at around 55–72 kDa present in crude plasma were removed in WGA-bound plasma fractions, while these proteins were major proteins found in WGA-unbound plasma fractions. However, there was no difference in the protein patterns from the plasma of CRC patients and healthy controls. The original gels and their densitometry analysis were shown in the Appendix A.

### 3.3. Label-free Relative Quantification of Plasma Glycoproteins

Label-free proteomic analysis was performed in three groups of WGA-bound plasma fractions from pooled plasma of non-metastatic, metastatic CRC patients and healthy controls. As shown in Figure 3A, 189 proteins were identified and compared. After filtering, 1227 accepted peptides which corresponded to 80 proteins were selected for protein quantification (Appendix A). Of these, only 62 proteins containing at least two peptides were considered (Appendix A). The relative expression levels of the 62 glycoproteins in non-metastatic and metastatic CRC patients were compared to those of healthy controls using volcano plots in Figure 3B,C, respectively. Using criteria of *p*-value less than 0.05 and expression levels greater than 1.5 fold change, three glycoproteins were up-regulated (red dots) and four glycoproteins were down-regulated (green dots) in the non-metastatic CRC patient group compared to those in the healthy controls, respectively. Using the same criteria, seven glycoproteins were up-regulated (red dots) and eight glycoproteins were down-regulated (green dots) in the metastatic CRC patient group compared to those of the healthy controls, respectively. The lists of 20 WGA-enriched glycoproteins showing differentially expressed levels >1.5-fold change and *p*-value < 0.05 in either non-metastatic or metastatic CRC patients compared to those of the healthy controls were shown in Table 2. Among these, complement C9 (C9) was the only glycoprotein significantly increased in both non-metastatic and metastatic CRC states, while only fibronectin (FN1) was significantly decreased in both CRC states.

In addition, N-linked glycosylation and glycation modifications were searched and identified against the protein sequences of these 20 proteins. We found that four glycoproteins including alpha-2 macroglobulin (A2M), haptoglobin (HP), alpha-1-acid glycoprotein 1 (ORM1) and complement C4-A (C4A) contained various types of N-linked glycosylation and one glycoprotein, A2M, contained glycated peptides (Table 2). These detected N-linked glycoforms contained N-acetylneuraminic acid (NeuAc), a sugar in complex glycosylation forms, which can interact with WGA resin during the enrichment preparation. Although we did not quantify the levels of these glycopeptide forms in each group, some N-liked glycoforms including HexNAc(4)Hex(5)NeuAc(2), HexNAc(3)Hex(6)Fuc(1)NeuAc(1), Hex(5)Hex(6)NeuAc(3), HexNAc(5)Hex(6)NeuAc(1), HexNAc(5)Hex(6)NeuAc(3), and HexNAc(4)Hex(5)NeuAc(2) were obviously detected in CRC patient groups. Moreover, we also detected three glycated sites of A2M with carboxymethyllysine (CML). The full details of N-linked and glycated sites and modification forms of these glycoproteins were shown in the Appendix A.

### 3.4. Protein-Protein Interaction (PPI) Network Analysis of Plasma Glycoproteins in CRC Patients

In order to evaluate functional interactions among identified glycoproteins in plasma of CRC patients, PPI network and biological interactions were mapped and constructed from glycoproteins showing expression changes with *p*-value < 0.05 in non-metastatic CRC and metastatic CRC compared to the control group, respectively. The PPI results showed that 35 nodes were obtained in non-metastatic CRC patients (Figure 4A), while 41 nodes were found in metastatic CRC patients (Figure 4B). Of these, 30 identified glycoproteins were shared in both CRC states, indicating that they may be associated with cancer progression. Interestingly, five glycoproteins, namely AZGP1, ITH1, ITH2, SERPINC1 and SERPING2 were significantly expressed only in plasma samples of CRC patients with non-metastasis, while 11 glycoproteins consisting of A1BG, A2M, C2, CFB, CPN2, HP, ORM1, SERPINA1, SERPINA6, TTR, and VTN were significantly expressed only in the patients with metastasis. The PPI network analysis revealed that most of the plasma glycoproteins enriched by WGA in both CRC groups were associated with three clusters of biological processes, including immune responses; complement pathways; wound healing and coagulation.

### 3.5. Verification of Biomaker Cadidates Identified by Quantitative Proteomics

According to label-free quantification results, C9 and FN1, which showed significantly up and down-regulation in CRC patients respectively, were selected to confirm their expression levels. The levels of C9 and FN1 in non-metastatic and metastatic CRC patient groups, as well as the healthy control group, were determined in WGA-bound plasma fraction and in crude plasma using immunoblot analysis (Figure 5). The immunoblot results revealed two bands at 72 kDa and above 55 kDa corresponding to C9 protein; however, the top band was rarely observed in the crude plasma samples due to presence of serum albumin while it was clearly seen in the WGA bound plasma fraction samples. FN1 was appeared in only one band above 250 kDa in the immunoblots. Based on band intensities, C9 level was elevated while FN1 level was decreased in non-metastatic and metastatic CRC groups in both WGA bound plasma fraction and crude plasma samples. Both glycoproteins were undetectable in the WGA-unbound plasma fraction samples. The original western blots of C9 and FN1 and their densitometry analysis were shown in the Appendix A.

### 3.6. Confirmation of C9 and FN1 Levels in the Study Cohort

To obtain a better resolution on the selected biomarker candidates in the study cohort, the expression level of C9 and FN1 of 30 individual crude plasma samples were determined by immunoblotting. Scatter plot of C9 immunoblots revealed a significant increase of C9 level in non-metastatic CRC patients (*p*-value = 0.0002, Figure 6A) and metastatic CRC patients (*p*-value = 0.0007) compared to those in the healthy controls. Of note, although the C9 level tended to be higher in metastatic CRC patients than in non-metastatic CRC patients, there was no statistically significant difference between the two groups (*p*-value = 0.0887). Scatter plot of FN1 immunoblots showed that FN1 level was significantly lower in both non-metastatic CRC patients (*p*-value = 0.0003, Figure 6B) and metastatic CRC patients (*p*-value = 0.0002) compared to those in the controls. Whereas FN1 plasma level was not difference between the two CRC states (*p*-value = 0.9570). The original western blots of C9 and FN1 and total proteins of all samples in the study cohort were shown in the Appendix A.

In order to compare the potential performance of C9 and FN1 with the routinely used biomarker CEA for detecting CRC, the CEA level of the same plasma samples were determined using ELISA. The level of CEA in the plasma samples of non-metastatic CRC patients (*p*-value = 0.0147, Figure 6C) and metastatic CRC patients (*p*-value = 0.0089) were significantly higher than those in the controls. But when compared between two CRC states, there was no significant difference (*p*-value = 0.4285). Of note, CEA level varied broadly in individual samples, especially in the two CRC stages, while C9 and FN1 level were present in a narrow range among all samples.

### 3.7. Validation of C9 and FN1 Expression in an Independent Cohort

To validate reproducibility of these differential findings, the levels of C9, FN1 and CEA were determined in crude serum samples in an another (validation) cohort by the same approaches used in the study cohort. The distributions of C9, FN1 and CEA levels in CRC (non-metastasis and metastasis) as well as healthy controls were showed and compared as depicted in Figure 6D,E, and Figure 6F, respectively. The result displayed significant increase in C9 levels in the serum samples of metastatic CRC patients when compared to those of non-metastatic CRC patients (*p*-value = 0.0047) and those of healthy controls (*p*-value < 0.0001), and there was no difference of C9 serum level between non-metastatic CRC and healthy controls (*p*-value = 0.3560). While, the results of FN1 from the study cohort and validation cohort showed good concordance. The level of FN1 was decreased in both non-metastatic CRC patients (*p*-value = 0.0227) and metastatic CRC patients (*p*-value = 0.0195) in comparison with healthy controls. Notably, when compared between two CRC states, the FN1 level was not significantly different (*p*-value = 0.7984). For CEA, its level in the patients with metastatic CRC was significantly increased as compared to those in the patients with non-metastatic CRC (*p*-value < 0.0001) and healthy controls (*p*-value = 0.0140), but there was not different between the patients with non-metastatic CRC and the controls (*p*-value = 0.0907). The original western blots of C9 and FN1 and total proteins of all samples in the independent cohort were shown in the Appendix A.

### 3.8. Evaluation of Biomarker Candidate Performance

To investigate the diagnostic performance of C9 and FN1 for CRC detection, ROC analysis was performed for C9, FN1, and CEA. The area under the ROC curve (AUC), sensitivity and specificity of each biomarker for CRC detection in both study and validation cohorts were shown in Figure 7. In discriminating between CRC patients and healthy controls, the overall accuracy of the biomarkers was represented by the AUC. In the study cohort, the AUC of C9 and FN1 were 0.93 and 0.95, respectively, which was higher than that of CEA (AUC = 0.83). The typical CEA cutoff value of 5 ng/mL used in the clinical diagnostics can distinguish CRC patients from the healthy controls with 40% sensitivity at 90% specificity. According to the ROC analysis, setting the C9 cut off level of >1.23 can discriminate healthy controls from CRC patients with the best combination of sensitivity (90%) and specificity (90%). Similarly, FN1 at cut off level of <0.49 enabled to discriminate the healthy controls from the CRC patients with sensitivity (90%) and specificity (90%).

To improve biomarker performance for diagnosing CRC in the current clinical practice, logistic regression on raw values of C9 and FN1 to establish a model with the combination of those two biomarker candidates was performed. C9 with a combination of FN1 yielded an increased predictive accuracy with AUC value of 0.99, the two-biomarker combination showed 100% sensitivity for CRC prediction at 89% specificity. Interestingly, a three-biomarker panel consisting of CEA, C9, and FN1 demonstrated an ideal performance in discriminating between CRC patients and healthy controls with AUC value of 1.00 and provided 100% predictive accuracy.

To validate the findings above, similar ROC analysis in the validation cohort was performed. The AUC of C9 and FN1 were 0.83 and 0.82, respectively which was lower than that of CEA (AUC = 0.88). However, the combination of C9 and FN1 can increase the AUC value to 0.89 which was slightly higher than CEA alone (AUC = 0.88). Moreover, three-biomarker panel consisting of CEA, C9, and FN1 enabled to increase AUC value up to 0.91 and improve the predictive accuracy for CRC.

## 4. Discussion

Different types of lectins have different specificity and affinity for glycoproteins in biological samples and provide different protein identification profiles [13,14]. Hagerbaumer et al. reported that WGA-bearing glycoproteins were increased in CRC tissues when compared to those in normal colorectal tissues [12]. Patwa et al. used Concanavalin (Con A) to enrich glycoproteins and found that plasma samples of patients with CRC or adenomas had dramatically higher levels of sialylation and fucosylation as compared to healthy controls [15]. Kim et al. demonstrated that 26 serum glycoproteins captured by phytohemagglutinin-L_4_ (L-PHA) were candidate CRC biomarkers [16]. Hence, we studied the WGA-enriched plasma glycoprotein profiles of CRC patient and healthy control samples. Using proteomic approaches and label-free relative quantitation, 62 glycoproteins were identified and compared among CRC patient groups and the healthy control group. Of these, 20 glycoproteins showed significantly differential expression levels between the two CRC stages and the healthy control group including three up-regulated and four down-regulated glycoproteins in the non-metastatic CRC patient group, and seven up-regulated and eight down-regulated glycoproteins in the metastatic CRC patient group (Figure 3 and Table 2). Based on the PPI network and biological interaction analysis, these proteins were predominantly involved in immune responses, complement pathways, wound healing and coagulation (Figure 4). Interestingly, complement C9 (C9) is the only glycoprotein showing a significant increase in plasma of non-metastatic and metastatic CRC patients, while only fibronectin (FN1) was significantly decreased in both CRC patient groups.

Complement C9 is one of five component proteins (C5b, C6, C7, C8 and C9) comprising of a membrane attack complex (MAC) which is a terminal event in the complement cascades. The MAC attaches to the surface of target cells and forms pores across the cell membrane resulting in complement-dependent cytotoxicity (CDC). The complement system plays important roles in innate and adaptive immune responses, providing an efficient immune surveillance and homeostasis [17]. However, it has been reported that the expression of complement proteins was elevated in malignant tumors and that complement activation in tumor microenvironment promotes tumorigenesis and progression [18]. Chong et al. reported C9 level was elevated significantly in plasma of gastric cancer patients when compared to that in healthy control groups detected by a label-free proteomics and C9 immunoblotting [19]. They also showed that C9 level was up-regulated in gastric cancer tissues and cell lines, suggesting that its increase may contribute in tumorigenesis. In addition, Cheng et al. showed, in a rat model, the up-regulation of C9 gene expression in esophageal adenocarcinoma compared with non-cancer epithelial cells [20]. According to our results, C9 level was up-regulated in plasma of CRC patient groups. Thus, complement C9 seems to be a promising biomarker candidate used for gastrointestinal cancer. However, the precise role of complements in carcinogenesis has not been fully established.

Fibronectin (FN1) is a glycoprotein which presented mainly in a soluble form found in the blood and also found in an insoluble form in the extracellular matrix (ECM) of tissues. FN1 plays important roles in cell adhesion, growth, migration as well as differentiation, which are mediated through integrin signaling. It has also been implicated in cancer-associated processes by promoting tumor growth, invasion, and metastasis [21,22]. In this study, FN1 was found to be depleted in plasma of patients with non-metastatic and metastatic CRC patients. The level of plasma FN1 varied in different cancers. Depleted plasma FN1 levels were observed in chronic lymphocytic leukemia and osteomyelosclerosis [23], whereas elevated plasma FN1 levels were found in breast cancer [24] and gastric cancer [25]. The role of FN1 in tumorigenesis and malignant progression has been highly controversial [26]. Taylor et al. reported that FN1 level was decreased in human and mouse tumor cell lines and its decrease was correlated with Met/HGF-mediated tumorigenesis, suggesting that FN1 acts as a tumor suppressive role [27]. On the contrary, evidence suggests that FN1 level was increased in many types of cancer [28,29,30]. The conclusions have been variable, probably because patients have not always been classified in terms of the precise origin of neoplastic tissues and FN1 level in cancer patients may be associated to complicating conditions such as inflammation. Therefore, further investigation is needed to find the precise role of FN1 in tumorigenesis, especially in CRC.

In this study, C9 and FN1 were chosen to be confirmed and validated, since the C9 level was increased and FN1 level was decreased in both CRC states compared with those of the healthy controls. Compared to quantitative proteomic results, immunoblotting revealed that C9 and FN1 levels were altered in similar manner in both study and validation cohorts (Figure 5 and Figure 6). Although the C9 level was likely to be higher in metastatic CRC patients than that in non-metastatic CRC patients, a larger cohort is needed to clarify whether it plays roles in the CRC development. Oppositely, the FN1 level was likely to be lower in both CRC states, indicating that the decease of FN1 level may be not related with the progression of CRC. Nevertheless, the combined detection of CEA, a commercial CRC biomarker, together with FN1 and CEA in crude plasma/serum samples showed an improvement of CRC diagnostic performance when compared to that detected by CEA level alone (Figure 7).

Other than C9 and FN1, among 20 significantly differential expressed glycoproteins, the presence of albumin in our study is of interest since albumin is widely known to be a non-glycosylated protein. We attempted to find the glycosylated and glycated albumin’s peptides in our mass spectrometry results using Byonic software, but found no one. Albumin is the most abundant protein ranged from 35–50 mg/mL in blood serum of an average adult [31]. It is a transport protein with capability of binding to a large variety of ligands such as fatty acids, metals, amino acids, and pharmaceutical compounds [31,32]. Albumin was previously reported to be statistically significant decreased in the late-stage stages of ovarian cancers and the serum albumin level was associated with poor survival rate [33]. Thus, decreasing of albumin level in CRC patients in our study was likely due to non-specific binding and co-elution with other WGA-binding proteins.

Along with albumin, all other glycoproteins in the Table 2 were searched for their possible glycosylated- and glycated-peptides. Even though our mass spectrometry isn’t the best choice for N-linked glycosylation study since we didn’t perform fragmentation in higher-energy collisional dissociation (HCD) mode, we could find N-linked glycosylation of some glycoproteins including alpha-2-macroglobulin (A2M; 9 glycoforms), haptoglobin (HP; 8 glycoforms), alpha-1-acid glycoprotein 1 (A1AG1 or ORM1; 2 glycoforms) and complement C4-A (CO4A or C4A; 1 glycoform) (Appendix A). In addition, three glycations were found in only A2M (Appendix A). A2M was the only glycoprotein that we could find its glycated peptides; this probably due to the fact that it is the most abundant glycoprotein in our study (168 peptide counts), thus it has more chance to be fragmented and be identified in our experimental set up. In addition, significant decrease of plasma A2M level in our study is in accordance with a previous study of prostate cancer [34]. However, the plasma level of A2M in other cancer patients can be varied, it can be increased [35] or unchanged [36] as well. Therefore, the value of A2M as cancer biomarker is still controversial.

Increasing of plasma HP level in our study, especially in metastatic CRC patients is also in agreement a previous study that demonstrate the potential of HP as a novel biomarker predicting CRC hepatic metastasis [37]. It has been reported to has aberrant glycosylation in patients with gastric cancer [38]. Haptoglobin in our study showed difference in glycosylation forms among the three study groups as well (Appendix A). Study of a protein glycosylation profiles and its glycosylation level is another interesting aspect that may benefit for novel biomarker discovery.

A1AG1 has a number of biological functions. It has the ability to modulate immune response and plays important roles in tumor microenvironment, cancer progression and metastasis. Alterations in the plasma A1AG1 level have been well documented in several types of cancer such as liver, breast cancer, lung, laryngeal, ovary, urothelial carcinoma and malignant mesothelioma [39]. In this study, we found an increase level of A1AG1 in the metastatic CRC patient group compared to the healthy control group. Therefore, A1AG1 may be another potential biomarker for common cancer. However, further study in comparison with other diseases is needed.

## 5. Conclusions

Here, we have developed a mass spectrometry-based proteomic approach in combination with the WGA glycoprotein enrichment to study differentially expressed plasma glycoproteins in non-metastatic and metastatic CRC patients compared to age-matched healthy controls. A number of glycoproteins were identified and compared between CRC patient and healthy control groups. C9 was the only glycoprotein showing a significant increase whereas FN1 was the only glycoprotein displaying a significant decrease in the plasma of both non-metastatic and metastatic CRC patients. The combination of CEA with FN1 and CEA in crude plasma samples showed an improvement of CRC diagnostic performance when compared to that detected by CEA level alone. However, due to the relatively small sample size used in this study, further validation of C9 and FN1 in a large-scale independent cohort and their roles in CRC carcinogenesis are needed to be examined and investigated. Nevertheless, other differentially expressed plasma glycoproteins identified in this study are also of interest for further validation. Moreover, some aberrant glycoforms predominant detected in A2M, HP, A1AG1, and C4-A in CRC patients are attracting attention. However, other MS/MS methods such as multiple reaction monitoring (MRM) analyzed by a triple quadrupole (QqQ) mass spectrometer, is needed to confirm these alterations. Finally, this study demonstrates the value of the glycoproteomic approach for identification of novel CRC biomarker candidates.

## Figures and Tables

**Figure 1 proteomes-08-00026-f001:**
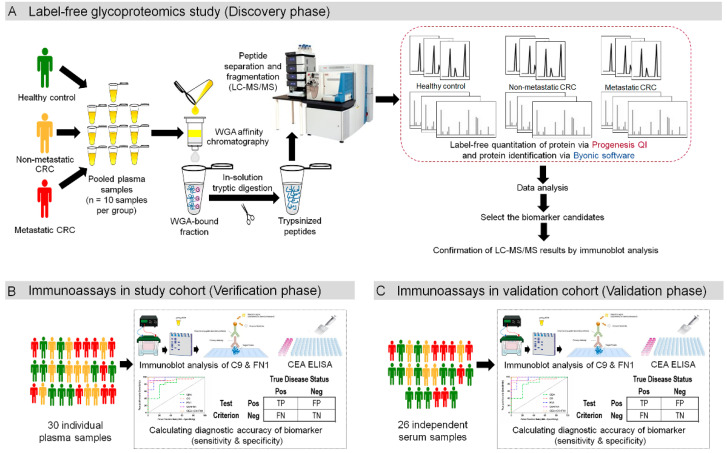
Selection of biomarker candidates and statistical analysis. (**A**) Discovery phase for biomarker candidates. Plasma samples of each group (healthy control, non-metastatic and metastatic CRC groups) were pooled and enriched by WGA kit. The WGA-enriched samples were tryptic-digested and analyzed by LC-MS/MS. Label free quantification of proteins were performed using Progenesis QI and the proteins were identified by Byonic search engine. Data analysis was performed to select the biomarker candidates. (**B**) Verification phase of biomarker candidates. The levels of the selected biomarker candidates and CEA were performed in individual plasma samples in the study cohort. ROC analysis and diagnostic accuracy parameters (sensitivity and specificity) were determined for the biomarkers. (**C**) Validation phase of biomarker candidates. The selected biomarker candidates were validated in an independent (validation) cohort and the statistical analysis was performed.

**Figure 2 proteomes-08-00026-f002:**
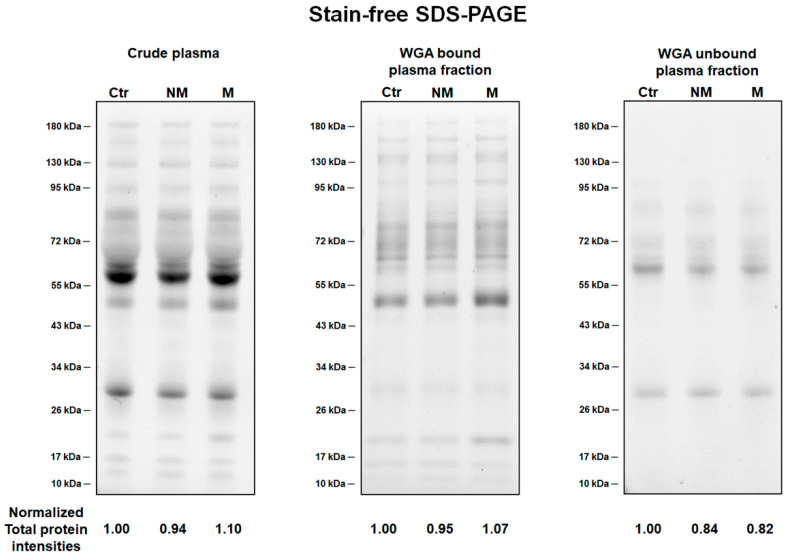
Protein patterns from pooled plasma samples of healthy control and CRC patient groups before and after WGA enrichment. Representative stain-free SDS-PAGE gels demonstrated the total protein patterns of pooled plasma of healthy controls (Ctr), non-metastatic CRC patients (NM), and metastatic CRC patients (M). Crude serum (10 µg), WGA-bound serum fraction (5 µg), and WGA-unbound serum fraction (5 µg) of three groups separated on 10% Stain-free SDS-PAGE. Total proteins were visualized by a stain-free gel system. Values below the gels denote the intensity of total protein bands for each sample normalized to those of the healthy control groups.

**Figure 3 proteomes-08-00026-f003:**
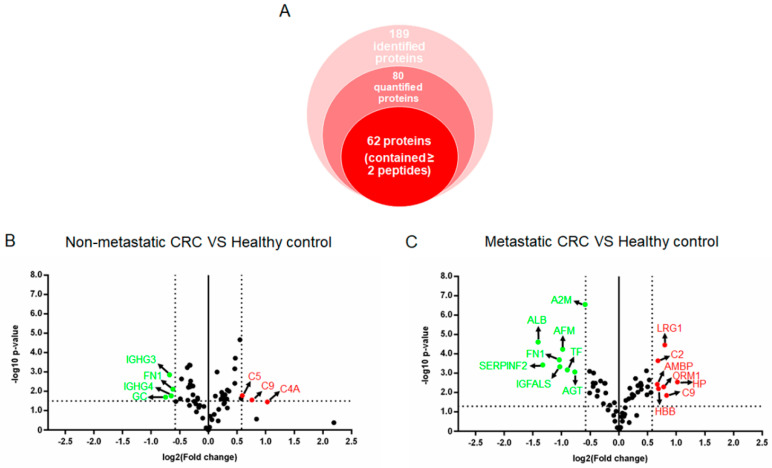
Proteins identified from WGA-enriched plasma of CRC patients. (**A**) The total number of identified proteins obtained from healthy controls and CRC patients (non-metastatic and metastatic CRC) by label-free relative quantification. (**B**,**C**) Volcano plots of the protein expressions in non-metastatic and metastatic CRC patients in comparison to those of the healthy controls, respectively. The x-axis represents log2 fold changes of proteins and the y-axis represents −log10 *p*-values. The red and green dots indicate proteins with significantly different expression of ≥1.5 and ≤−1.5 fold-change and *p*-values < 0.05, respectively. The black dots indicate proteins which were not significantly altered between CRC patients and healthy control groups.

**Figure 4 proteomes-08-00026-f004:**
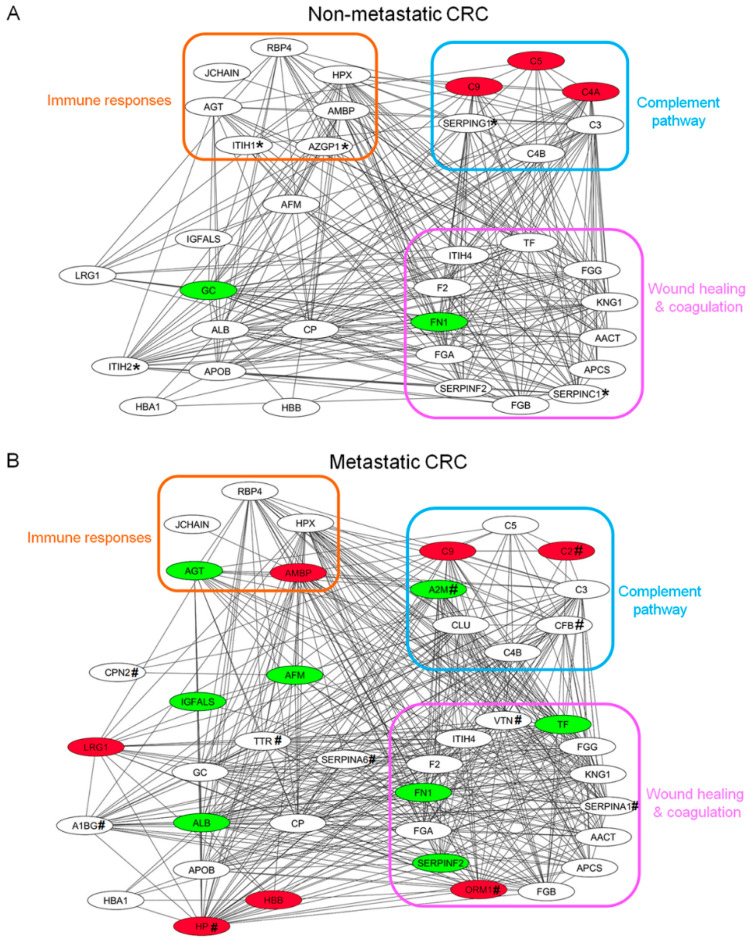
The protein-protein interaction (PPI) network of significantly expressed plasma glycoproteins in CRC patients. (**A**) The PPI of non-metastatic CRC consisted of 35 nodes. (**B**) The PPI of metastatic CRC constructed showing 41 nodes. The PPI was performed by STRING and represented by Cytoscape software. Each node lists the gene name of the identified proteins. (red nodes, up-regulated ≥1.5 fold; green nodes, down-regulated ≤−1.5 fold; white nodes, <±1.5 fold). * indicated glycoproteins significantly expressed only in non-metastatic CRC patients and # indicated glycoproteins significantly expressed only in metastatic CRC patients.

**Figure 5 proteomes-08-00026-f005:**
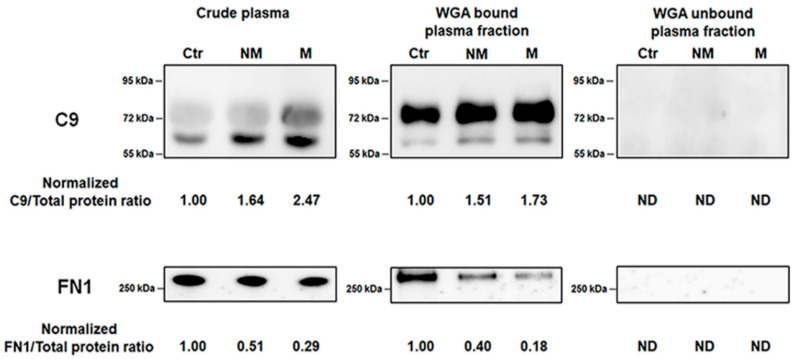
Immunoblots and the expression levels of complement C9 (C9) and fibronectin (FN1) of crude plasma, WGA-bound plasma and WGA-unbound plasma fractions in heathy control and CRC patient groups of the study cohort. Representative immunoblots of C9 and FN1 from three types of pooled samples, including crude plasma (10 µg), WGA-bound plasma fraction (5 µg), and WGA-unbound plasma fraction (5 µg). Each sample type was loaded on gels and subsequently immunoblotted with anti-complement C9 or anti-fibronectin. Values below immunoblots denote the ratio of C9 or FN1 divided by its total protein band intensities for each sample and the normalized ratio compared to those of the healthy control groups. ND indicated undetectable.

**Figure 6 proteomes-08-00026-f006:**
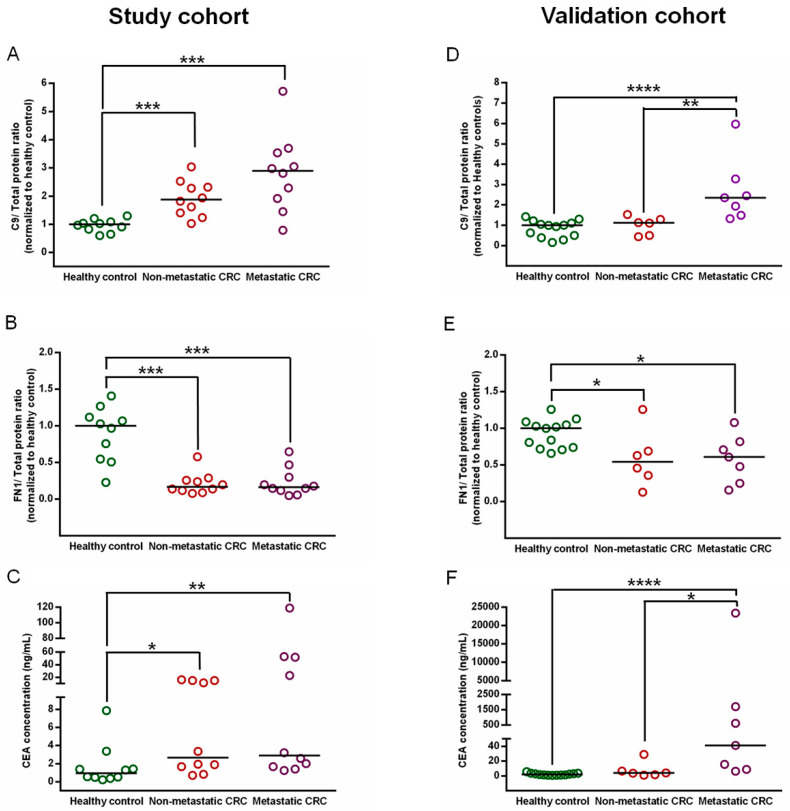
Scatter plots display of biomarker distributions for healthy control, non-metastatic CRC, and metastatic CRC in the study cohort and validation cohort. (**A**,**B**) Scatter plots of the relative ratios of C9/total proteins and FN1/total proteins detected by immunoblotting in crude plasma samples of non-metastatic CRC patients (*n* = 10) and metastatic CRC patients (*n* = 10), compared to median of those in the healthy controls (*n* =10), respectively. (**C**) Scatter plot of the CEA level detected by ELISA in plasma samples of all subjects in the study cohort. (**D**,**E**) Scatter plots of the relative ratios of C9/total proteins and FN1/total proteins detected by immunoblotting in crude serum samples of non-metastatic CRC patients (*n* = 6) and metastatic CRC patients (*n* = 7), compared to median of those in the healthy controls (*n* =13), respectively. (**F**) Scatter plot of the CEA level detected by ELISA in serum samples of all subjects in the validation cohort. * represents *p*-value < 0.05, ** represents *p*-value < 0.01, *** represents *p*-value < 0.001, and **** represents *p*-value < 0.0001.

**Figure 7 proteomes-08-00026-f007:**
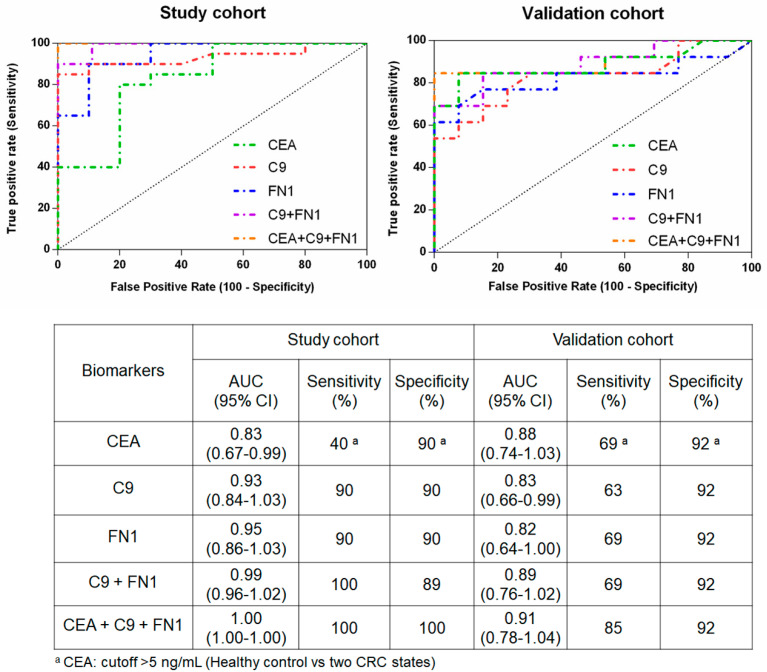
Performance of CRC biomarker candidates, C9 and FN1, and CEA. ROC curves (top) were showed for the study and validation cohorts. AUC, sensitivity and specificity as determined for each biomarker candidate and their combination.

**Table 1 proteomes-08-00026-t001:** Characteristics of CRC patients and healthy controls used in the study and validation cohorts.

Description	Study Cohort (*n* = 30)			Validation Cohort (*n* = 26)		
	Healthy Control	Non-Metastatic CRC	Metastatic CRC	Healthy Control	Non-Metastatic CRC	Metastatic CRC
***N***	10	10	10	13	6	7
**Gender**						
Female	4	4	4	8	5	6
Male	6	6	6	5	1	2
**Median age (range)**	51 (40–66)	58 (41–78)	59 (41–72)	56 (50–63)	59 (33–66)	60 (56–64)
**Histopathology**						
Well-differentiated adenocarcinoma		4	2		1	3
Moderately differentiated adenocarcinoma		6	4		5	4
Poorly differentiated adenocarcinoma		0	4		0	0

**Table 2 proteomes-08-00026-t002:** List of 20 WGA-enriched plasma proteins with a 1.5-fold change cutoff in differential expression at *p*-value less than 0.05 in non-metastatic or metastatic CRC patients compared to those of the healthy controls.

No.	UniProt	Gene	Description	Peptides Counts *	Non-Metastatic CRC vs. Healthy Control		Metastatic CRC vs. Healthy Control	
					*p*-Value	Fold Change (NM/Ctr)	*p*-Value	Fold Change (M/Ctr)
1	sp|P00738	HP	Haptoglobin **	60	>0.05	1.07	0.0028	2.02
2	sp|P02748	C9	Complement component C9	4	0.0278	1.69	0.0138	1.78
3	sp|P02750	LRG1	Leucine-rich alpha-2-glycoprotein	14	0.0098	1.28	<0.0001	1.74
4	sp|P02763	ORM1	Alpha-1-acid glycoprotein 1 **	14	>0.05	1.26	0.0050	1.71
5	sp|P68871	HBB	Hemoglobin subunit beta	15	0.0257	1.25	0.0063	1.61
6	sp|P06681	C2	Complement C2	2	>0.05	−1.03	0.0002	1.60
7	sp|P02760	AMBP	Protein AMBP	10	0.0007	1.38	0.0037	1.60
8	sp|P02774	GC	Vitamin D-binding protein	2	0.0198	−1.69	0.0097	1.47
9	sp|P01031	C5	Complement C5	6	0.0165	1.51	0.0208	1.12
10	sp|P0C0L4	C4A	Complement C4-A **	5	0.0350	2.05	>0.05	−1.06
11	sp|P01861	IGHG4	Immunoglobulin heavy constant gamma 4	5	0.0169	−1.57	>0.05	−1.14
12	sp|P01860	IGHG3	Immunoglobulin heavy constant gamma 3	3	0.0014	−1.61	0.0370	−1.21
13	sp|P01023	A2M	Alpha-2-macroglobulin **^,^***	168	>0.05	1.00	<0.0001	−1.51
14	sp|P01019	AGT	Angiotensinogen	7	0.0288	−1.29	0.0008	−1.71
15	sp|P02787	TF	Serotransferrin	2	0.0061	−1.28	0.0007	−1.87
16	sp|P43652	AFM	Afamin	19	0.0029	−1.24	0.0001	−1.98
17	sp|P35858	IGFALS	Insulin-like growth factor-binding protein complex acid labile subunit	6	0.0049	−1.23	0.0005	−2.05
18	sp|P02751	FN1	Fibronectin	5	0.0077	−1.54	0.0002	−2.05
19	sp|P08697	SERPINF2	Alpha-2-antiplasmin	4	0.0151	−1.21	0.0004	−2.52
20	sp|P02768	ALB	Serum albumin	63	0.0333	−1.49	<0.0001	−2.66

* Number of peptides used for quantification, NM indicated non-metastatic CRC, M indicated metastatic CRC, and Ctr indicated healthy control. ** proteins contained N-linked glycosylation, *** proteins contained glycated modifications.

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
