# Peer review of "Glycoproteomic Analysis Reveals Aberrant Expression of Complement C9 and Fibronectin in the Plasma of Patients with Colorectal Cancer"

_proteomes, 2020, doi:10.3390/proteomes8030026_

Round 1

Reviewer 1 Report

Submitted manuscript is interesting, original and present new data; however there are some criticism to be underlined.

Authors collected 30 samples, including 20 from patients, but unfortunately them were pooled and one sample per group was available. It should be better to analyze in duplicate the single patients

Because the manuscript concerns glycosylation, title should be focused on it.

Authors did not distinguish complex glycosylation, such as FN1 and Complement from non-enzymatic one, such as for albumin and hemoglobin; in fact, non-enzymatic glycosylation may be identified by MS/MS spectra

Few proteins were considered, and authors need to extend discussion at more proteins.

In table 2 are reported 20 interesting proteins, but authors validated and discussed complement C9, only; other proteins are to be validated and discussed.

In addition, fold changes between metastatic and non-metastatic CRC are to be allowed

PPI investigation is interesting, but the number of proteins was not sufficient for an extensive analysis of PPI; 4 proteins for metastatic CRC

Minor comments

Figure 2 may be deleted because it is not informative

Figure 4 is not informative, eventually it should be report all shared proteins in the two panels.

At line 91 “…plasma samples …“ is it mg or mL?

Author Response

Summary: We thank the reviewers for their helpful and constructive comments.  Additions or modifications made in direct response to the Reviewer’s comments are marked by the track changes of Microsoft office. Responses to specific issues are listed below in the order in which they appeared in the Reviewer’s critique. Changes made in response to those comments are also indicated by reference to the appropriate line numbers in the main text.

Referee: 1

  1. Authors collected 30 samples, including 20 from patients, but unfortunately them were pooled and one sample per group was available. It should be better to analyze in duplicate the single patients

Answer: Thank you for the reviewer’s comment. We divided samples into 3 groups (10 samples/group; healthy controls, non-metastatic and metastatic CRC patients). For the quantitative proteomic analysis, we performed LC-MS/MS of the pooled sample of each group in triplicate runs. Therefore, nine runs were used for protein quantitation and identification. Although the results obtained quantitative proteomic analysis came from the pooled one sample of each group, we confirmed the expression levels of some proteins i.e. complement C9 (C9) and fibronectin (FN1), from the pooled one sample of each group by westernblotting (Fig. 5). Both results are consistent. We added another cohort to validate these results. The levels of C9 and FN1 of the study and validation cohorts were shown in similar direction (Fig.6). We believe that one pooled sample per group is still reliable and could be a representative of each group.

We agree with the reviewer that analyzing in duplicate the single patients would be better; however, it will create more variation in protein profiling of each single samples that are likely to be hard and challenging for all peptide mapping alignments by the software (Progenesis QI). In addition, a super computer is needed to analyze the larger multiple LC-MS/MS runs alignments. Although our technique may not be the best, it allows us to find the differential expression levels of certain proteins among these groups.    

We added additional information for clarification in the text line 97-100. In addition, we added another cohort to study the levels of C9 and FN1 in the methods (line 87-91, 95) and the result (Fig. 6) and discussion and conclusion. 

  1. Because the manuscript concerns glycosylation, title should be focused on it.

Answer: We modified a title to be related the glycoproteomic approach we used. The edited title is shown in the text line 2-4.

  1. Authors did not distinguish complex glycosylation, such as FN1 and Complement from non-enzymatic one, such as for albumin and hemoglobin; in fact, non-enzymatic glycosylation may be identified by MS/MS spectra.

Answer: Thank you for the reviewer’s comment. We extensively searched the MS/MS data by Byonic with 148 forms of N-linked glycosylation as well as 15 forms of glycation (non-enzymatic glycosylation) against 20 identified proteins shown in Table 2. Predominant glycoforms associated with acetylneuraminic acid were obviously detected in alpha-2 macroglobulin, haptoglobin, alpha-1-acid glycoprotein 1, and complement C4-A of CRC patient groups. We showed these results in the text line 169-182 and 282-294 as well as in the supplementary Table S4 and S5.  

  1. Few proteins were considered, and authors need to extend discussion at more proteins.

Answer: We added more proteins for discussion in the text line 494-529.

  1. In table 2 are reported 20 interesting proteins, but authors validated and discussed complement C9, only; other proteins are to be validated and discussed.

Answer: We examined the expression level of C9 in the study cohort (n=30) and the validation cohort (n=26). In addition, we examined the level of FN1 in two cohorts.  

  1. In addition, fold changes between metastatic and non-metastatic CRC are to be allowed

Answer: We added details of proteins expressed between metastatic and non-metastatic CRC patients in the text line 317-322.

  1. PPI investigation is interesting, but the number of proteins was not sufficient for an extensive analysis of PPI; 4 proteins for metastatic CRC

Answer: We added details of identified proteins in figure 4.

  1. Figure 2 may be deleted because it is not informative

Answer: We would like to keep this figure.

  1. Figure 4 is not informative, eventually it should be report all shared proteins in the two panels.

Answer: We added details of identified proteins in figure 4.

  1. At line 91 “…plasma samples …“ is it mg or mL?

Answer: It is “mg of protein”.

Reviewer 2 Report

  1. Figure 6: C9 and CEA levels in plasma of heathy control and CRC patient groups should be validated in more cases, but not only in the 10 non-metastatic CRC patients and 10 metastatic CRC patients.
  2. The differentially expressed proteins between non-metastatic CRC patients and metastatic CRC patients should be listed. Furthermore, the rationale of these proteins as prognostic biomarkers for CRC metastasis and survival should be also investigated.
  3. Please address the scientific and clincial significance of the findings in Figure 4.

Author Response

Summary: We thank the reviewers for their helpful and constructive comments.  Additions or modifications made in direct response to the Reviewer’s comments are marked by the track changes of Microsoft office. Responses to specific issues are listed below (green text) in the order in which they appeared in the Reviewer’s critique. Changes made in response to those comments are also indicated by reference to the appropriate line numbers in the main text.

Referee: 2

  1. Figure 6: C9 and CEA levels in plasma of heathy control and CRC patient groups should be validated in more cases, but not only in the 10 non-metastatic CRC patients and 10 metastatic CRC patients.

Answer: We added another cohort set (n=26) of healthy controls and CRC patient groups for validation. The C9 and CEA levels of the study and validation cohorts were shown in Figure 6. In addition, we determined and validated FN1 level in individual samples of two cohorts (Fig. 6). The levels of C9 and FN1 were up- and down-regulated in CRC patient groups, respectively.      

  1. The differentially expressed proteins between non-metastatic CRC patients and metastatic CRC patients should be listed. Furthermore, the rationale of these proteins as prognostic biomarkers for CRC metastasis and survival should be also investigated.

Answer: We explained more details of the differentially expressed proteins between non-metastatic and metastatic CRC patients in the text line 279-281 and 317-322.

  1. Please address the scientific and clinical significance of the findings in Figure 4.

Answer: We modified figure 4 to be more informative.

Round 2

Reviewer 2 Report

The authors have addressed my comments.